# Latent Reasoning VLA: Latent Thinking and Prediction for Vision-Language-Action Models

**Shuanghao Bai** [* 1 2] **Jing Lyu** [* 2 3 4] **Wanqi Zhou** [1] **Zhe Li** [2] **Dakai Wang** [1] **Lei Xing** [1] **Xiaoguang Zhao** [3]
**Pengwei Wang** [2] **Zhongyuan Wang** [2] **Cheng Chi** [2] **Badong Chen** [1] **Shanghang Zhang** [2 5]

🌐 Project Page   ⬡ Code

## Abstract

Vision-Language-Action (VLA) models benefit from chain-of-thought (CoT) reasoning, but existing approaches incur high inference overhead and rely on discrete reasoning representations that mismatch continuous perception and control. We propose Latent Reasoning VLA (**LaRA-VLA**), a unified VLA framework that internalizes multimodal CoT reasoning into continuous latent representations for embodied action. LaRA-VLA performs unified reasoning and prediction in latent space, eliminating explicit CoT generation at inference time and enabling efficient, action-oriented control. To realize latent embodied reasoning, we introduce a curriculum-based training paradigm that progressively transitions from explicit textual and visual CoT supervision to latent reasoning, and finally adapts latent reasoning dynamics to condition action generation. We construct two structured CoT datasets and evaluate LaRA-VLA on both simulation benchmarks and long-horizon real-robot manipulation tasks. Experimental results show that LaRA-VLA consistently outperforms state-of-the-art VLA methods while reducing inference latency by up to 90% compared to explicit CoT-based approaches, demonstrating latent reasoning as an effective and efficient paradigm for real-time embodied control.

---

[*]Equal contribution   [1]Institute of Artificial Intelligence and Robotics, Xi'an Jiaotong University. [2]Beijing Academy of Artificial Intelligence. [3]Institute of Automation, University of Chinese Academy of Sciences. [4]School of Artificial Intelligence, University of Chinese Academy of Sciences. [5]State Key Laboratory of Multimedia Information Processing, School of Computer Science, Peking University. Correspondence to: Cheng Chi <chicheng@baai.ac.cn>, Badong Chen <chenbd@mail.xjtu.edu.cn>, Shanghang Zhang <shanghang@pku.edu.cn>.

*Proceedings of the $43^{rd}$ International Conference on Machine Learning*, Seoul, South Korea. PMLR 306, 2026. Copyright 2026 by the author(s).

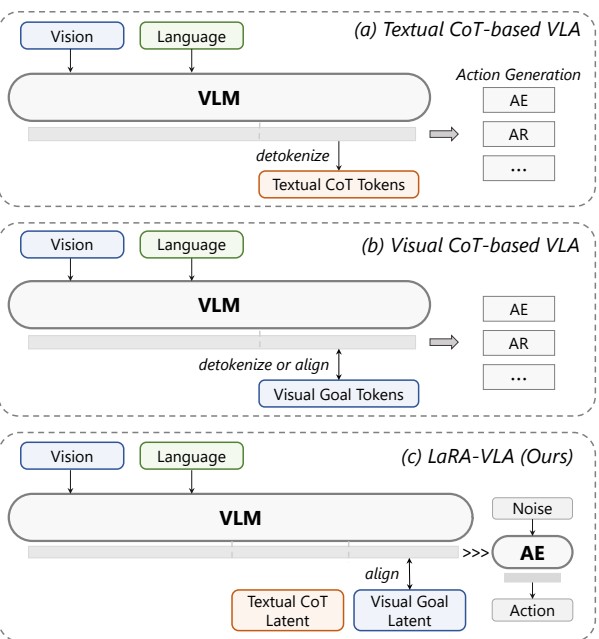

*Figure 1.* Comparison of CoT formulations in VLA models. (a) Textual CoT-based VLA explicitly generates discrete reasoning tokens and decodes them into actions through autoregressive (AR) policies or action experts (AE). (b) Most visual CoT-based VLA represents reasoning as discrete visual goal tokens before action generation. (c) Our model internalizes textual and visual reasoning into continuous latents. Visual goal latents align with perceptual features, encode future-oriented task information, and provide implicit supervision for textual CoT latents.

## 1. Introduction

Vision-Language-Action (VLA) models have emerged as a promising direction for scalable, general-purpose robotic manipulation (Kim et al., 2025b; Bai et al., 2025b), as they aim to end-to-end map rich multimodal observations and language instructions to continuous control commands. To improve VLA performance, a variety of training and inference strategies have been proposed (Bai et al., 2025c; Sridhar et al., 2025; Lin et al., 2025b). Among these approaches, incorporating Chain-of-Thought (CoT) reasoning

---

has proven particularly effective (Chen et al., 2025c). CoT is commonly framed as a teaching signal that encourages structured intermediate reasoning, thereby improving interpretability and generalization in long-horizon robotic tasks.

Existing CoT approaches for VLA models can be broadly categorized by modality, as shown in Figure 1. Text-based CoT methods represent intermediate reasoning explicitly in natural language, covering task decomposition and high-level planning, and may additionally verbalize information derived from visual observations (Zawalski et al., 2025; Sun et al., 2025b; Fan et al., 2025; Tan et al., 2026). Visual CoT methods instead express reasoning through visual reconstruction or prediction, typically by modeling future observations or intermediate visual states (Zhao et al., 2025a; Zhang et al., 2025b). A third line of work combines textual and visual CoT, leveraging multi-modal intermediate representations for reasoning (Zhang et al., 2025a).

Despite their effectiveness, existing CoT-based methods face two fundamental challenges. First, text-based CoT often requires extensive textual reasoning during inference, leading to substantial computational overhead. Long reasoning traces dramatically increase token length, resulting in excessive KV-cache usage, high memory consumption, and increased latency. In practice, such models may operate at control frequencies below 5 Hz, or even around 1 Hz (Zawalski et al., 2025), which is unacceptable for real-time robotic control. Second, most CoT formulations rely on discrete representations: textual CoT is constrained to language tokens (Zawalski et al., 2025; Lin et al., 2025a), while visual CoT is typically aligned with discrete visual tokens produced by VQ-based tokenizers (Zhao et al., 2025a). We argue that CoT is effective not because it is expressed in natural language, but because it exposes structured intermediate reasoning. In embodied settings, where both perception and action evolve in continuous spaces, constraining reasoning to discrete tokens introduces a representational mismatch between reasoning and control.

To address these challenges, we propose Latent Reasoning VLA (LaRA-VLA), a unified latent-reasoning VLA framework that performs reasoning and prediction entirely in latent space for robotic action. A comparison between LaRA-VLA and existing CoT-based VLA methods is summarized in Table 1. Inspired by latent chain-of-thought reasoning methods (Hao et al., 2024; Pham & Ngo, 2025; Huang et al., 2025), we adapt latent reasoning to embodied VLA models, enabling structured reasoning to be internalized into continuous latent representations. Different from Fast-ThinkAct (Huang et al., 2025), which mainly distills textual CoT into latent representations while retaining visually grounded reasoning as discrete traces, LaRA-VLA internalizes both textual and future-oriented visual CoT into continuous latents. The visual goal latents further serve as

implicit multimodal supervision for textual latent reasoning, making the learned latent dynamics directly useful for action generation.

Our training paradigm follows a three-stage progression that transitions from explicit multi-modal reasoning to latent embodied reasoning. Training begins with unified textual and visual CoT supervision, where textual reasoning steps and future visual predictions are explicitly aligned with their corresponding annotations. We then adopt a curriculum-based strategy that models embodied reasoning as a sequence of continuous latent states, gradually internalizing reasoning structure by replacing explicit textual CoT with a compact set of latent representations and relying on visual prediction objectives as implicit supervision. Once textual CoT is fully absorbed into latent reasoning, the model is further adapted to action generation, enabling latent reasoning dynamics to directly condition continuous control outputs. Throughout training, visual latents used for supervision are encoded by the same visual encoder as the input observations, and an exponential moving average (EMA) encoder is employed as a target network to stabilize latent representation learning and prevent representation collapse.

Overall, our approach extends latent CoT reasoning from language-only settings to embodied VLA models, enabling efficient, action-oriented reasoning without explicit CoT generation at inference time. By coupling latent reasoning dynamics directly with action generation, LaRA-VLA allocates computation to compact latent "thought steps" rather than verbose textual reasoning, substantially reducing token expansion and inference latency. This makes the approach well-suited for real-time robotic control. Moreover, representing reasoning as continuous latent dynamics avoids the representational mismatch introduced by discrete language or visual tokens. To support this paradigm, we construct an automated CoT annotation pipeline that provides structured supervision, including subtask decomposition, movement reasoning, and target object localization, and curate latent reasoning datasets across both simulated and real-world environments, including LIBERO-LaRA, Bridge-LaRA, and real-robot demonstrations. Our contributions are threefold:

- We introduce a latent-reasoning paradigm for Vision–Language–Action models, in which chain-of-thought reasoning is internalized into continuous latent representations across textual and visual modalities, enabling inference-efficient reasoning that aligns with continuous perception and control.

- We propose LaRA-VLA, a unified VLA model that realizes this paradigm through a curriculum-based training strategy, progressively transitioning from explicit multi-modal CoT supervision to latent embodied reasoning, guided by predictive visual latent objectives and stabilized with EMA-based visual encoders.

*Table 1.* A taxonomy of VLA models based on the representation forms of chain-of-thought CoT and actions. Specifically, we categorize models by whether their textual CoT is represented as explicit discrete tokens or continuous latent states, whether their visual CoT aligns with raw pixels, discrete visual tokens, or encoded continuous visual latents, and whether the action output is represented as discrete tokens or continuous values.

| Method | Venue | Text CoT | | Visual CoT | | Action |
| --- | --- | --- | --- | --- | --- | --- |
| | | Presence | Form | Presence | Align Form | Form |
| *VLAs with Text CoT* | | | | | | |
| ECoT (Zawalski et al., 2025) | CoRL 2024 | ✓ | Discrete Token | ✗ | – | Discrete |
| GraspVLA (Deng et al., 2025) | CoRL 2025 | ✓ | Discrete Token | ✗ | – | Continuous |
| $\pi_{0.5}$ (Intelligence et al., 2025) | CoRL 2025 | ✓ | Discrete Token | ✗ | – | Continuous |
| ThinkAct (Huang et al., 2025) | NeurIPS 2025 | ✓ | Discrete Token | ✗ | – | Continuous |
| Fast-ThinkAct (Huang et al., 2026) | CVPR 2026 | ✓ | Discrete Token & Continuous Latent | ✗ | – | Continuous |
| *VLAs with Visual CoT* | | | | | | |
| CoT-VLA (Zhao et al., 2025a) | CVPR 2025 | ✗ | – | ✓ | Discrete Visual Tokens | Discrete |
| DreamVLA (Zhang et al., 2025b) | NeurIPS 2025 | ✗ | – | ✓ | Discrete Visual Tokens | Continuous |
| UD-VLA (Chen et al., 2025b) | ICLR 2026 | ✗ | – | ✓ | Discrete Visual Tokens | Discrete |
| VITA (Ma et al., 2025) | arXiv 2025 | ✗ | – | ✓ | Discrete Visual Tokens | Discrete |
| *VLAs with Both Text and Visual CoT* | | | | | | |
| UP-VLA (Zhang et al., 2025a) | ICML 2025 | ✓ | Discrete Token | ✓ | Discrete Visual Tokens | Discrete |
| **LaRA-VLA (Ours)** | – | ✓ | Continuous Latent | ✓ | Continuous Latent | Continuous |

- We construct two structured chain-of-thought datasets, LIBERO-LaRA and Bridge-LaRA, featuring multimodal reasoning annotations for embodied manipulation, and conduct extensive evaluations in both simulation and long-horizon real-robot tasks to demonstrate the effectiveness and robustness of LaRA-VLA.

**Conflict of Interest Disclosure**   Several authors are employed by or affiliated with Beijing Academy of Artificial Intelligence (BAAI), which conducts research on foundation models and embodied AI systems related to this work.

## 2. Related Work

### 2.1. Vision Language Action Models

VLA models extend large multimodal language models to robotic control, with extensive efforts devoted to architectural advances (Black et al., 2024; Intelligence et al., 2025; Kim et al., 2025a; Cui et al., 2025; Li et al., 2025b;c; 2024b) as well as training- and inference-time optimizations (Sridhar et al., 2025; Lin et al., 2025b; Bai et al., 2026) since the introduction of RT-2 (Zitkovich et al., 2023). Among these directions, CoT reasoning at training time has proven particularly effective. As summarized in Table 1, existing CoT-based VLA methods can be broadly categorized into textual CoT and visual CoT. Textual CoT methods generate explicit textual reasoning traces to refine task instructions or extract motion- and object-relevant information from visual observations (Zawalski et al., 2025; Sun et al., 2025b; Huang et al., 2025). Visual CoT methods, in contrast, reconstruct or predict visual observations conditioned on multimodal inputs, typically relying on discrete visual tokens produced by

VQ-VAE-based representations (Zhao et al., 2025a; Zhang et al., 2025b; Lv et al., 2025). Despite their effectiveness, both paradigms rely on discrete tokenized reasoning. Textual CoT incurs long autoregressive generation chains with high inference cost, while both textual and visual CoT exhibit a mismatch with the inherently continuous perception and action spaces in robotics. To address these limitations, we introduce latent reasoning, in which textual CoT is replaced by compact continuous latent variables learned via curriculum-style training, and visual CoT is aligned with continuous visual representations from the perception backbone.

A closely related work is Fast-ThinkAct (Huang et al., 2025), which reduces the inference cost of VLA CoT reasoning by distilling explicit reasoning traces into latent representations. However, it mainly latentizes textual CoT, while visually grounded reasoning remains in a discrete trace-based form. In contrast, LaRA-VLA progressively internalizes both textual CoT and future-oriented visual CoT into continuous latent representations, where visual goal latents provide implicit supervision for textual latent reasoning and directly support action generation.

### 2.2. Latent Reasoning in LLMs and VLMs

Explicit CoT can improve reasoning performance but often incurs verbose intermediate outputs, high inference latency, and a reliance on discrete tokens. These drawbacks have motivated implicit or continuous reasoning in latent space, which preserves multi-step computation without explicit reasoning traces (Xu et al., 2025; Ruan et al., 2025). In language models, latent reasoning is typically instantiated

through hidden states or continuous variables that function as implicit thought. For instance, Coconut (Hao et al., 2024) shows that latent reasoning supports richer internal search while achieving a more favorable accuracy–efficiency trade-off than explicit CoT. SIM-CoT (Wei et al., 2025) identifies optimization instability when scaling implicit reasoning tokens and introduces supervised stabilization, while CoDi (Shen et al., 2025) distills explicit CoT into continuous latent representations via self-distillation. Latent reasoning has also been extended to vision–language models, where multimodal latents serve as internal reasoning states rather than explicit symbolic traces (Sun et al., 2025a; Pham & Ngo, 2025). To adapt latent reasoning to VLA, we adopt a staged training strategy that initializes structured reasoning with explicit CoT and progressively transfers it into latent space. Unlike prior methods that rely solely on answer supervision, we jointly ground latent reasoning in visual representations and action signals, enabling control-relevant reasoning that supports stable policy learning and efficient action generation without explicit CoT at inference time.

## 3. Method

In this section, we present the complete pipeline of our Latent Reasoning VLA (LaRA-VLA) framework. We first describe the construction of structured CoT datasets in Section 3.1. We then introduce the model architecture of LaRA-VLA and detail its training procedures in Sections 3.2 and 3.3, respectively.

### 3.1. Data Collection

Effective robotic manipulation requires the joint modeling of long-horizon subtask structure, spatial grounding of target objects, and motion-level reasoning for action execution. However, existing data collection pipelines typically capture these components in isolation, leading to redundant or incomplete supervision (e.g., exhaustive object-level bounding boxes in ECoT (Zawalski et al., 2025) or missing target localization in Emma-x (Sun et al., 2025b)). To address these limitations, we develop a fully automated annotation pipeline following an anchor-first, generate-later paradigm driven by semantic and temporal anchors. Semantic anchors are extracted using Qwen3-VL (Bai et al., 2025a) from the initial frame and task instruction to identify the manipulated object, while temporal anchors are obtained by segmenting robot trajectories into atomic manipulation stages based on gripper state changes. Conditioned on these anchors, Qwen3-VL generates concise subtask descriptions, and open-vocabulary grounding with GroundingDINO (Liu et al., 2024) and SAM3 (Carion et al., 2025) produces temporally consistent target object bounding boxes. Motion reasoning is derived from end-effector trajectories by computing goal-directed and local motions, which are discretized

into directional descriptors and incorporated into the CoT annotations. Details are provided in Figure 10 and Appendix B.

Based on this pipeline, we construct two benchmark datasets on simulated environments, LIBERO (Liu et al., 2023) and SimplerEnv (Li et al., 2025a), resulting in *LIBERO-LaRA* and *Bridge-LaRA*, and further apply the same framework to a long-horizon real-world robotic manipulation dataset collected on physical hardware.

### 3.2. Model Architecture

We adopt Qwen3-VL (Bai et al., 2025a) as the backbone VLM to leverage its strong built-in reasoning capability, and directly inherit its image encoder to ensure consistent visual representations throughout training. To predict visual goal information, we introduce a dedicated <img_next> token to represent predicted visual latents, which enables explicit supervision and alignment during early-stage latent reasoning learning. For action prediction in Stages I and II, we follow an autoregressive action token design similar to (Pertsch et al., 2025), allowing the model to learn action generation jointly with latent reasoning in a stable and efficient manner. In the final stage, we remove explicit action token prediction and instead activate a dedicated action expert, decoupling action generation from token-level autoregressive decoding. Specifically, action generation is performed by a 16-layer Diffusion Transformer composed of alternating self-attention and cross-attention layers, which conditions on the learned latent representations to produce continuous action trajectories.

### 3.3. Training Procedures

**Stage I: Explicit CoT Fine-Tuning.** In the first stage, we fine-tune the VLM on embodied datasets with explicit CoT annotations constructed in Section 3.1. These annotations provide structured reasoning supervision, including task decomposition, movement reasoning, and target object information, enabling the model to adapt to embodied manipulation tasks with explicit intermediate reasoning. Given input images and a language instruction, the image encoder first maps the visual observation to a sequence of visual tokens, denoted as $\mathbf{v}$, while the instruction text is tokenized into textual tokens, denoted as $\mathbf{x}$. The VLM jointly attends to both visual and textual tokens and is trained to autoregressively generate a sequence of CoT tokens. During this stage, discrete ground-truth textual tokens are used to explicitly supervise CoT generation via teacher forcing. The training objective is defined as the negative log-likelihood of the ground-truth CoT sequence:

$$\mathcal{L}_{\text{cot}} = -\sum_{t=1}^{T_{\text{CoT}}} \log p_\theta(c_t \mid c_{<t}, \mathbf{v}, \mathbf{x}), \tag{1}$$

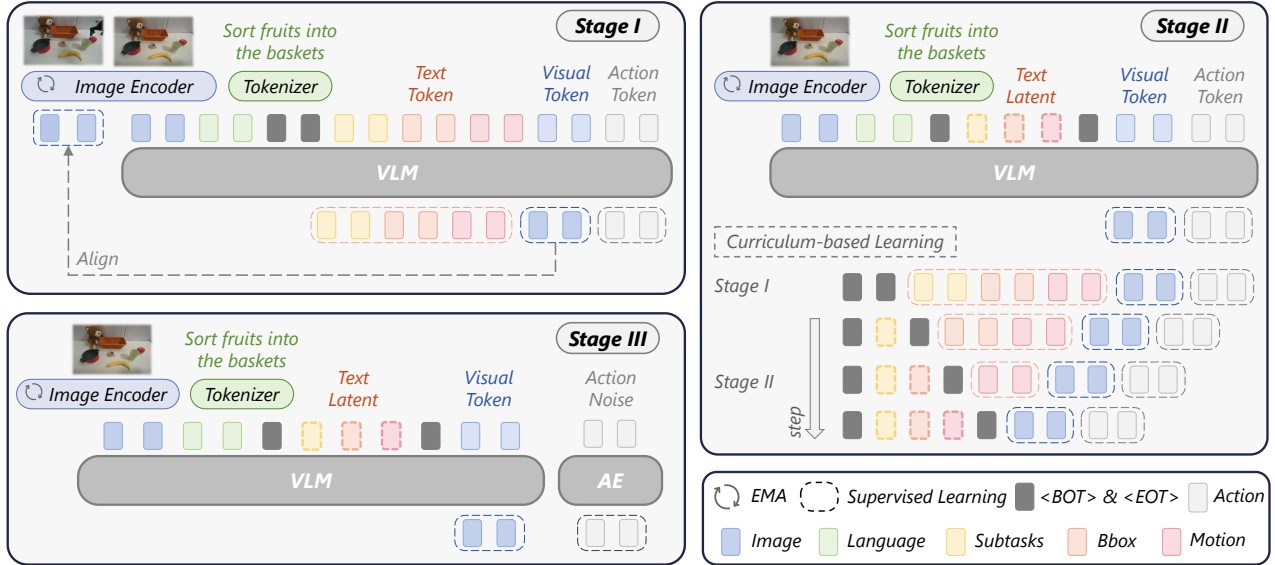

*Figure 2.* **Overview of LaRA-VLA.** Training proceeds in three stages: (i) explicit CoT fine-tuning with aligned visual prediction latents and inverse-dynamics supervision for actions; (ii) a curriculum-based transition from explicit CoT to compact text latents, gradually reducing the number of text tokens while increasing reliance on latent reasoning, where the latent representations are also implicitly supervised by visual and action signals; and (iii) adaptation of latent-conditioned VLM features to an action expert for efficient action generation without explicit CoT at inference time.

where $\{c_t\}_{t=1}^{T_{\text{CoT}}}$ is the ground-truth CoT token sequence, $c_{<t}$ is the preceding CoT tokens, and $p_\theta(\cdot)$ is the conditional token distribution parameterized by the VLM.

In addition to textual supervision, we align visual reasoning by predicting future visual latents. Let $\mathbf{z}_{t+1}$ denote the visual latent representation of the next observation encoded by the same visual encoder used for the input observations. The VLM predicts this latent from the current context, yielding the following alignment objective:

$$\mathcal{L}_{\text{vis}} = \|\hat{\mathbf{z}}_{t+1} - \mathbf{z}_{t+1}\|_1, \tag{2}$$

where $\hat{\mathbf{z}}_{t+1}$ is the predicted visual latent and $\mathbf{z}_{t+1}$ is obtained by encoding the next observation. To stabilize visual latent learning and prevent representation collapse, we follow prior work (Chen et al., 2025a) and update the parameters used to compute target visual latents using an exponential moving average (EMA) of the online encoder parameters:

$$\bar{\theta}_v^t = \tau_v \, \bar{\theta}_v^{t-1} + (1 - \tau_v) \, \theta_v^t, \tag{3}$$

where $\theta_v^t$ denotes the parameters of the online visual encoder at iteration $t$, $\bar{\theta}_v^t$ denotes the corresponding EMA-averaged parameters used to compute stable target visual latents, and $\tau_v$ is the decay rate.

We further leverage the predicted future visual representations together with the preceding visual and textual context to infer actions via an inverse dynamics model. Specifically, we employ an inverse dynamics function $f(\mathbf{v}_t, \mathbf{v}_{t+1} \mid \mathbf{x}, c) = \mathbf{a}_t$, which estimates the action that induces the

transition between consecutive visual states conditioned on the instruction and the intermediate reasoning step. To efficiently propagate action information across latent reasoning steps, we adopt the fast recursive generation framework of (Pertsch et al., 2025), which assigns coarse-grained action semantics to all latent representations. Concretely, action tokens are trained using an autoregressive objective, similar to Equation 1, yielding the action-token loss $\mathcal{L}_{\text{act-dis}}$.

**Stage II: Curriculum-based Replacement of Discrete CoT Tokens.** In the second stage, we internalize explicit textual reasoning into the latent space through a curriculum-based training strategy. Embodied reasoning is modeled as a sequence of continuous latent states, and discrete CoT tokens are progressively replaced by latent representations during training. Formally, we adopt the same training objectives as in Stage I, including the textual CoT likelihood and the visual latent prediction loss. However, instead of supervising all reasoning steps with discrete ground-truth CoT tokens, we gradually mask out subsets of CoT tokens and replace them with learnable latent representations. The proportion of discrete CoT tokens decreases over training according to a predefined curriculum schedule, until the entire chain-of-thought is fully internalized in the latent space.

**Stage III: Action Generation via Flow Matching.** To adapt latent reasoning to continuous control, we train the action prediction module using a flow matching objective. Let $\mathbf{a}_t$ denote the ground-truth action at time step $t$, and let $\epsilon \sim \mathcal{N}(\mathbf{0}, \mathbf{I})$ be Gaussian noise. Following the flow

*Table 2.* Performance comparisons with state-of-the-art methods on LIBERO, grouped by different CoT paradigms.

| CoT Type | Method | Spatial | Goal | Object | Long | Avg. |
|---|---|---|---|---|---|---|
| No CoT | OpenVLA (Kim et al., 2025b) | 84.7 | 88.4 | 79.2 | 53.7 | 76.5 |
| | $\pi_0$ (Black et al., 2024) | 96.8 | 98.8 | 95.8 | 85.2 | 94.2 |
| | OpenVLA-OFT (Kim et al., 2025a) | 97.6 | 98.4 | 97.9 | 94.5 | 97.1 |
| Textual CoT | ThinkAct (Huang et al., 2025) | 88.3 | 91.4 | 87.1 | 70.9 | 84.4 |
| | MolmoAct (Lee et al., 2025) | 87.0 | 95.4 | 87.6 | 77.2 | 86.6 |
| | $\pi_{0.5}$ (Intelligence et al., 2025) | 98.8 | 98.2 | 98.0 | 92.4 | 96.8 |
| | DeepThinkVLA (Yin et al., 2025) | 99.0 | 96.6 | 96.4 | 96.2 | 97.0 |
| Visual CoT | CoT-VLA (Zhao et al., 2025a) | 87.5 | 91.6 | 87.6 | 69.0 | 81.1 |
| | DreamVLA (Zhang et al., 2025b) | 97.5 | 94.0 | 89.5 | 89.5 | 92.6 |
| | F1 (Lv et al., 2025) | 98.2 | 97.8 | 95.4 | 91.3 | 95.7 |
| | UD-VLA (Chen et al., 2025b) | 94.1 | 95.7 | 91.2 | 89.6 | 92.7 |
| Latent CoT | Fast-ThinkAct (Huang et al., 2026) | 92.0 | 97.2 | 90.2 | 79.4 | 89.7 |
| | **LaRA-VLA (Ours)** | 96.4 | 98.6 | 99.8 | 96.6 | **97.9** |

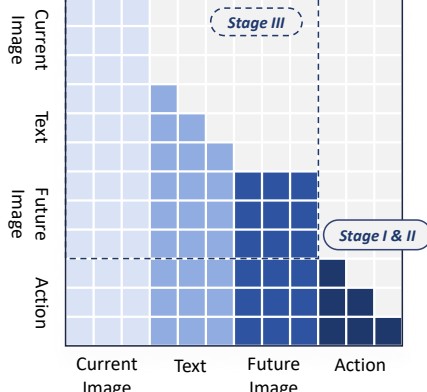

*Figure 3.* Attention mechanism used in LaRA-VLA.

matching formulation, we define a linear interpolation between noise and action as $\mathbf{a}_\tau = (1 - \tau)\epsilon + \tau\,\mathbf{a}_t$, where $\tau \sim \mathcal{U}(0, 1)$. The action expert predicts a velocity field $v_\theta(\mathbf{a}_\tau, \tau \mid \mathbf{h}_t)$ conditioned on a multi-modal latent context $\mathbf{h}_t$. Specifically, $\mathbf{h}_t$ aggregates the latent representation of the current visual observation and language instruction, the intermediate text-based reasoning latent, and the predicted future visual latent produced by the VLM. Through the inverse dynamics supervision applied in earlier stages, this latent context already encodes coarse-grained action-relevant information. As a result, we do not introduce an additional action latent, and actions are generated directly from the shared multi-modal latent representation. The flow matching loss is defined as

$$\mathcal{L}_{\text{act-con}} = \mathbb{E}_{\mathbf{a}_t, \epsilon, \tau} \left[ \|v_{\theta_a}(\mathbf{a}_\tau, \tau \mid \mathbf{h}_t) - (\mathbf{a}_t - \epsilon)\|_2^2 \right], \quad (4)$$

where $\mathbf{h}_t$ denotes the multi-modal latent context produced by the VLM at time step $t$.

**LaRA-VLA Attention Mechanism.** We introduce an attention mechanism tailored to our three-stage training paradigm, as illustrated in Figure 3. The model operates on multimodal tokens including text, current image, future image, and action tokens, with attention constraints explicitly regulating cross-token information flow. Here, text tokens serve as a unified abstraction that corresponds to language instructions and textual chain-of-thought in Stages I and II, and to text latents in Stages II and III. In Stages I and II, future image tokens attend causally to text and current image tokens, while interacting bidirectionally among themselves. Action tokens are generated autoregressively: each action token attends to all preceding text, current image, and future image tokens, as well as previously generated action tokens. In Stage III, action tokens are excluded from the attention computation, and the model is trained solely over text and vision tokens under the same attention constraints.

**Loss Function.** Our training objective is stage-dependent and follows a curriculum-style design. In Stage I, we jointly optimize the CoT supervision loss, the visual alignment loss, and the discrete action loss, i.e., $\mathcal{L}_{\text{cot}} + 0.1\mathcal{L}_{\text{vis}} + \mathcal{L}_{\text{act-dis}}$, to initialize structured reasoning and action grounding. In Stage II, we progressively anneal the explicit CoT supervision loss $\mathcal{L}_{\text{cot}}$ to zero. The model is lastly optimized using $0.2\,\mathcal{L}_{\text{vis}} + \mathcal{L}_{\text{act-dis}}$, which promotes latent-space reasoning while maintaining accurate action semantics. Finally, in Stage III, discrete action supervision is replaced by continuous action regression, and we optimize $\mathcal{L}_{\text{act-con}}$ to enable efficient continuous action generation.

## 4. Experiments

We evaluate the effectiveness of LaRA-VLA and the overall system through a comprehensive set of experiments spanning both simulation benchmarks and real-world robotic manipulation tasks. Our experiments are designed to address the following questions:

- How effective is LaRA-VLA compared to state-of-the-art methods in simulation benchmarks? (Section 4.1)

*Table 3.* Performance comparisons with state-of-the-art methods on SimplerEnv-WindowX, grouped by different CoT paradigms.

| CoT Type | Method | Put Spoon | Put Carrot | Stack Block | Put Eggplant | Avg. |
|---|---|---|---|---|---|---|
| No CoT | OpenVLA (Kim et al., 2025b) | 0.0 | 0.0 | 0.0 | 4.1 | 1.0 |
| | Octo (Ghosh et al., 2024) | 47.2 | 9.7 | 4.2 | 56.9 | 29.5 |
| | OpenVLA-OFT (Kim et al., 2025a) | 12.5 | 4.2 | 8.3 | 37.5 | 39.6 |
| | $\pi_0$ (Black et al., 2024) | 29.1 | 0.0 | 16.7 | 62.5 | 40.1 |
| | CogACT (Li et al., 2024a) | 71.7 | 50.8 | 15.0 | 67.5 | 51.3 |
| Textual CoT | ThinkAct (Huang et al., 2025) | 58.3 | 37.5 | 8.7 | 70.8 | 43.8 |
| Visual CoT | F1 (Lv et al., 2025) | 50.0 | 70.8 | 50.0 | 66.7 | 59.4 |
| | UD-VLA (Chen et al., 2025b) | 58.3 | 62.5 | 54.1 | 75.0 | 62.5 |
| Latent CoT | **LaRA-VLA (Ours)** | 95.8 | 62.5 | 25.0 | 91.7 | **68.8** |

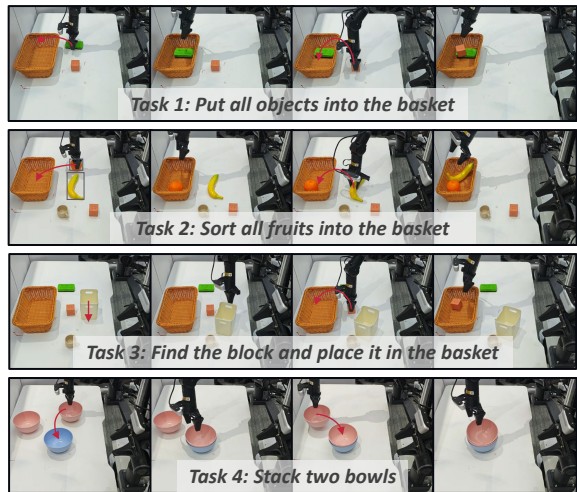

*Figure 4.* Real-world setup of four long-horizon tasks.

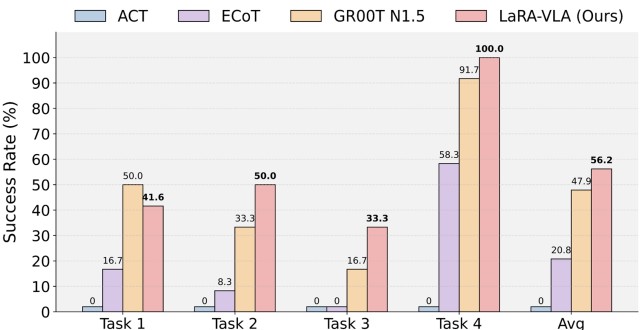

*Figure 5.* Real-world results.

• How well does LaRA-VLA perform on **long-horizon** real-world manipulation tasks compared to state-of-the-art approaches? (Section 4.2)

• How effective are the latent reasoning components in LaRA-VLA, and what additional advantages does our approach offer? (Section 4.3)

### 4.1. Simulation Experiments

**Benchmarks and Datasets.** We conduct experiments on two widely used benchmarks, LIBERO (Liu et al., 2023) and SimplerEnv (Li et al., 2025a). LIBERO consists of four task suites, Spatial, Goal, Object, and Long, each containing 10 single-arm manipulation tasks. We report success rates for each suite and the overall average over 50 rollouts per task. SimplerEnv evaluates real-to-sim generalization of robot manipulation policies trained on real-world data. We evaluate on WidowX robots across four tasks and report per-task success rates and the overall average over 24 rollouts per task. Based on these benchmarks, we construct two training datasets, LIBERO-LaRA and Bridge-LaRA, which are used to train LaRA-VLA.

**Baselines.** For LIBERO, we compare against a broad set of state-of-the-art VLA methods covering different CoT paradigms. No-CoT baselines include OpenVLA (Kim et al., 2025b), $\pi_0$ (Black et al., 2024), and OpenVLA-OFT (Kim et al., 2025a). Textual CoT baselines include ThinkAct (Huang et al., 2025), MolmoAct (Lee et al., 2025), $\pi_{0.5}$ (Intelligence et al., 2025), and DeepThinkVLA (Yin et al., 2025). Visual CoT methods include CoT-VLA (Zhao et al., 2025a), DreamVLA (Zhang et al., 2025b), F1 (Lv et al., 2025), and UD-VLA (Chen et al., 2025b). We additionally compare with the latent CoT method Fast-ThinkAct (Huang et al., 2026). For SimplerEnv, we follow the standard evaluation protocol and compare against representative baselines. No-CoT methods include Open-VLA, Octo (Ghosh et al., 2024), OpenVLA-OFT, $\pi_0$, and CogACT (Li et al., 2024a). We further include textual CoT (ThinkAct) and visual CoT (F1, UD-VLA) baselines. Implementation details and training hyperparameters of LaRA-VLA are provided in Appendix A.

**Results.** Tables 2 and 3 summarize quantitative results on the LIBERO and SimplerEnv-WidowX benchmarks. On LIBERO, LaRA-VLA achieves the best overall performance with an average success rate of 97.9%, including 99.8% on the Object suite and 96.6% on the Long suite, demonstrating strong object-centric reasoning and robustness in long-horizon manipulation. On SimplerEnv-WidowX, which evaluates real-to-sim generalization under diverse visual

*Table 4.* Robustness under visual perturbations. We report task success rates under Gaussian blur and Gaussian noise with two severity levels. H and L denote high- and low-severity perturbations, respectively.

| Benchmark | Method | Gaussian Blur-H | Gaussian Blur-L | Gaussian Noise-H | Gaussian Noise-L |
|---|---|---|---|---|---|
| LIBERO | Qwen-GR00T (Community, 2026) | 30.0 | 76.0 | 55.7 | 87.9 |
| | **LaRA-VLA (Ours)** | **42.9** | **79.4** | **76.0** | **92.7** |
| SimplerEnv | Qwen-GR00T (Community, 2026) | 13.5 | 35.4 | 8.3 | 22.9 |
| | **LaRA-VLA (Ours)** | **56.3** | **62.5** | **22.9** | **31.2** |

*Table 5.* Ablation study of different forms of CoT supervision on SimplerEnv. Text-CoT denotes explicit textual chain-of-thought, Latent Text-CoT denotes latent textual chain-of-thought, and Latent Vis-CoT denotes latent visual chain-of-thought.

| Text-CoT | Latent Text-CoT | Latent Vis-CoT | SR (%) |
|---|---|---|---|
| ✗ | ✗ | ✗ | 55.2 |
| ✓ | ✗ | ✗ | 58.3 |
| ✗ | ✓ | ✗ | 64.6 |
| ✗ | ✗ | ✓ | 63.5 |
| ✗ | ✓ | ✓ | **68.8** |

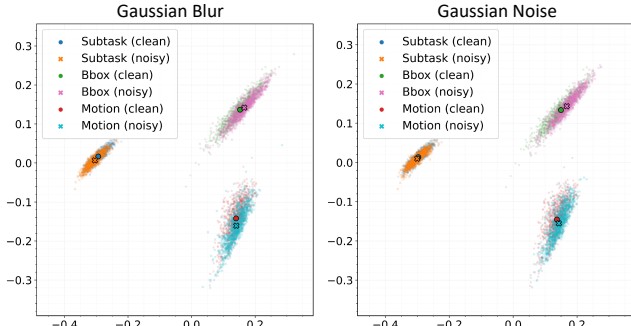

*Figure 6.* Latent-space distributions under Gaussian blur and Gaussian noise. Clean and perturbed latent remain clustered by semantic role, with only moderate distributional shifts, suggesting that the learned latent representations are stable under visual corruptions.

conditions, LaRA-VLA attains the highest average success rate of 68.8%, outperforming No-CoT, Textual CoT, and Visual CoT baselines. Across both benchmarks, LaRA-VLA consistently surpasses textual and visual CoT methods, indicating that latent reasoning provides more effective and stable guidance for action prediction and generalizes better than explicit CoT supervision.

## 4.2. Real-World Experiments

**Real-World Setup.** As illustrated in Figure 4, our real-world setup uses an Agilex Cobot Magic wheeled platform equipped with three RGB-D cameras. We consider four categories of long-horizon manipulation tasks: putting all objects into the basket, sorting fruits into the basket, finding a block and placing it into the basket, and stacking two bowls. For data collection, we record 100 demonstration trajectories per task category at 30 Hz. During evaluation, each task is executed for 12 rollout trials. We compare our method against ACT (Zhao et al., 2023), ECoT (Zawalski et al., 2025) and GR00T N1.5 (Bjorck et al., 2025) as baselines. Implementation details are provided in Appendix A.

**Results.** As shown in Figure 5, LaRA-VLA achieves the highest average success rate among all compared methods, substantially outperforming ACT and ECoT and surpassing GR00T N1.5 overall. In particular, LaRA-VLA attains the best performance on three out of four long-horizon real-world manipulation tasks, with especially notable gains on tasks that require multi-stage reasoning and sustained temporal coordination. These results suggest that latent reasoning improves robustness to error accumulation over long-horizon manipulation.

## 4.3. Analysis

**Ablation Study.** Table 5 shows that latent CoT supervision provides substantially larger gains than explicit textual CoT. While explicit textual CoT yields only marginal improvement over the no-CoT baseline, latent textual CoT leads to a significant increase in success rate, highlighting the effectiveness of internalized textual reasoning. Further incorporating latent visual CoT yields the best performance, as it both injects predictive visual information about future states and implicitly regularizes the latent textual CoT through multimodal alignment. Overall, these results indicate that structured latent CoT representations, especially when jointly learned across modalities, are more effective than explicit textual reasoning for embodied policy learning.

**Robustness of Lantent Representations and LaRA-VLA.** To further evaluate whether the learned latent representations remain stable under input perturbations, we conduct additional stress tests on LIBERO and SimplerEnv with two common visual corruptions: Gaussian noise and Gaussian blur. For Gaussian noise, we use standard deviations of $50$ and $30$ for the high- and low-severity settings, respectively. For Gaussian blur, we use a kernel size of $15 \times 15$ with $\sigma_x = \sigma_y = 5$ for the high-severity setting, and a kernel size of $9 \times 9$ with $\sigma_x = \sigma_y = 3$ for the low-severity setting.

We analyze both latent-space changes and task-level robustness. As shown in Figure 6, the latent reasoning tokens remain semantically consistent under corrupted observa-

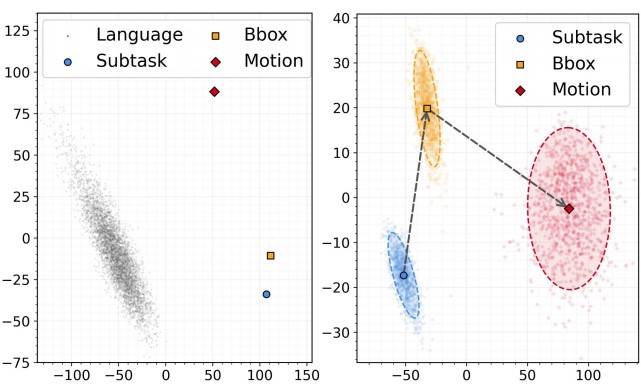

Figure 7. Latent collapse analysis.

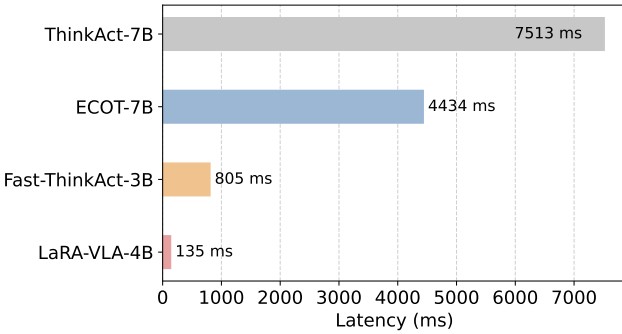

Figure 8. Inference time comparison on an NVIDIA A100 GPU.

tions, with only moderate distributional shifts. This suggests that the learned latent space does not collapse or become highly unstable under visual perturbations. We further compare LaRA-VLA with Qwen-GR00T (Community, 2026), which serves as a no-CoT baseline without latent reasoning. As shown in Table 4, both models degrade under visual corruption, but LaRA-VLA consistently maintains higher success rates across all perturbation types and severity levels. These results indicate that latent CoT reasoning not only improves clean performance, but also enhances robustness to corrupted visual inputs.

**Latent Collapse.** As shown in Figure 7, we observe no evidence of latent representation collapse. Latent tokens associated with different reasoning components form well-separated and semantically coherent clusters, demonstrating clear functional specialization rather than degeneration into uniform or uninformative representations. Moreover, latent representations of language instruction tokens (gray points) remain structured and occupy a distinct subspace from reasoning latents, indicating that latent CoT does not trivially reuse language embeddings. Together, these findings show that predictive supervision and action grounding provide sufficient inductive bias to maintain structured latent reasoning, even without explicit discrete chain-of-thought generation at inference time.

**Inference Efficiency.** As shown in Figure 8, LaRA-VLA achieves the lowest inference latency among all compared methods, requiring only 135 ms per rollout. Compared with explicit CoT methods, LaRA-VLA avoids autoregressive textual reasoning and reduces inference time by up to 90%. Notably, LaRA-VLA is also faster than Fast-ThinkAct, which uses latent CoT reasoning, suggesting that our compact latent design further improves efficiency beyond simply replacing explicit CoT with latent tokens. This advantage may come from using fewer latent reasoning tokens, compact visual goal latents, and a lightweight action expert.

## 5. Limitations

Although LaRA-VLA achieves fast inference and strong performance through latent chain-of-thought reasoning, several limitations remain and warrant further investigation. First, latent CoT representations may suffer from collapse in the absence of explicit supervision, where the semantics of latent tokens degenerate toward homogeneous representations, particularly as the number of latent tokens increases (Wei et al., 2025). To mitigate this risk, our current implementation restricts latent reasoning to a single token per step, which may limit expressiveness. Second, the training procedure is not maximally efficient. LaRA-VLA employs a curriculum learning strategy that gradually replaces explicit CoT tokens with latent representations. As training progresses, the number of CoT-related tokens increases, resulting in higher training cost. Improving training efficiency while preserving stable latent reasoning remains an important direction for future work.

## 6. Conclusion

We presented LaRA-VLA, a latent reasoning framework for Vision–Language–Action models that internalizes chain-of-thought reasoning into continuous latent representations across both textual and visual modalities. Rather than generating long explicit CoT sequences at inference time, LaRA-VLA replaces them with compact textual CoT latents and employs a curriculum-based training strategy to progressively transfer explicit reasoning into latent space. Visual latents are aligned with continuous perceptual features encoded by a shared visual encoder and stabilized using an exponential moving average, providing implicit supervisory signals that guide the learning of textual CoT latents. Experiments on simulated benchmarks and long-horizon real-robot manipulation tasks demonstrate that LaRA-VLA achieves strong performance while significantly improving inference efficiency, supporting the view that structured reasoning for embodied agents can be effectively realized in latent space without explicit chain-of-thought generation.

## Acknowledgements

This work was supported by the National Natural Science Foundation of China under Grant Nos. U25A20540, 62436005, and 62476011.

## Impact Statement

This paper aims to advance machine learning methods for robotic manipulation by improving the efficiency and scalability of reasoning in Vision–Language–Action models. The proposed approach introduces architectural and training innovations that enable effective reasoning without explicit chain-of-thought generation, potentially facilitating real-time robotic deployment. While such advances may benefit a wide range of robotic applications, including automation and assistive technologies, we do not identify any societal risks uniquely introduced by this work beyond those generally associated with robotic systems. As with related research, the broader impact will depend on how the technology is applied and governed in practice.

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

# A. Implementation Details

## A.1. Implementation Details of LaRA-VLA

**Training Paradigm.** We adopt the progressive three-stage training paradigm across all experiments. In **Stage I**, the input sequence encompasses explicit Chain-of-Thought (CoT) annotations, `<img_next>` tokens for next-frame feature prediction, and action tokens tokenized via fast (Pertsch et al., 2025) tokenizer. In **Stage II**, we facilitate the learning of implicit reasoning by substituting concrete CoT steps with `<thinking>` tokens. Finally, in **Stage III**, we discard explicit CoT supervision. The specific input formats for each stage are illustrated in Figure 9.

```
Stage I: Explicit Supervision
'Place the potato inside the bowl.  @ Subtask:  carry the potato toward the bowl.
BBox:  [0.5664 0.5898 0.6953 0.6641].  Reasoning:  the robot is closing the gripper.
<img_next> <img_next> <img_next> <img_next> <img_next> <img_next> <img_next> <img_next>
<img_next> <img_next> <img_next> <img_next> <img_next> <img_next> <img_next> <img_next>'

Stage II: Mixed Strategy (Transition)
One thinking token:
'Place the potato inside the bowl.  @ <|start_of_thinking|> <|thinking|>
<|end_of_thinking|> BBox:  [0.5664 0.5898 0.6953 0.6641].  Reasoning:  the robot is
closing the gripper.  <img_next> <img_next> <img_next> <img_next> <img_next> <img_next>
<img_next> <img_next> <img_next> <img_next> <img_next> <img_next> <img_next> <img_next>
<img_next> <img_next>'
Two thinking tokens:
'Place the potato inside the bowl.  @ <|start_of_thinking|> <|thinking|> <|thinking|>
<|end_of_thinking|> Reasoning:  the robot is closing the gripper.  <img_next> <img_next>
<img_next> <img_next> <img_next> <img_next> <img_next> <img_next> <img_next> <img_next>
<img_next> <img_next> <img_next> <img_next> <img_next> <img_next>'
Three thinking tokens:
'Place the potato inside the bowl.  @ <|start_of_thinking|> <|thinking|> <|thinking|>
<|thinking|> <|end_of_thinking|> <img_next> <img_next> <img_next> <img_next> <img_next>
<img_next> <img_next> <img_next> <img_next> <img_next> <img_next> <img_next> <img_next>
<img_next> <img_next> <img_next>'

Stage III: Fully Latent Reasoning
'Place the potato inside the bowl.  @ <|start_of_thinking|> <|thinking|> <|thinking|>
<|thinking|> <|end_of_thinking|> <img_next> <img_next> <img_next> <img_next> <img_next>
<img_next> <img_next> <img_next> <img_next> <img_next> <img_next> <img_next> <img_next>
<img_next> <img_next> <img_next>'
```

*Figure 9.* Example of training data formats.

**Training Config.** Most hyperparameters are shared across evaluation settings, with only a small subset adapted to each scenario. In particular, the action horizon is set to 16 for the Bridge dataset, 8 for the LIBERO benchmark, and 25 for real-world experiments. Detailed hyperparameter configurations for the LIBERO, Bridge, and real-world evaluations are summarized in Table 6. All models are trained using 8 NVIDIA H100 GPUs.

## A.2. Implementation Details Baselines in Real-world Experiments

**ACT (Zhao et al., 2023).** We instantiate ACT using the LeRobot implementation, configured with a chunk size of $K = 50$ and $n_{\text{action}} = 50$ action steps per chunk. The perception stack follows the original design, using a ResNet-18 vision backbone pretrained on ImageNet and mean–std normalization for visual, state, and action inputs. The transformer backbone employs a 4-layer encoder and a 1-layer decoder with model dimension 512, 8 attention heads, and a 3200-dimensional feedforward network with ReLU activation and dropout 0.1. Following prior work, we enable the VAE module with a 32-dimensional latent space, 4 encoder layers, and a KL weight of 10.0. Training is performed for 40k gradient steps with batch size 100 on a single H100 GPU, using Adam with learning rate $1 \times 10^{-5}$, weight decay $1 \times 10^{-4}$, and the same learning rate for the backbone. Note that we adopt the LeRobot fix for the ACT decoder, which uses a single effective decoder layer consistent with the original implementation.

**GR00T N1.5 (Bjorck et al., 2025).** We use the original GR00T N1.5 implementation with its default architecture and a

*Table 6.* Hyperparameter settings in LaRA-VLA across LIBERO, SimplerEnv, and real robot experiments.

| Hyperparameters | LIBERO | | | SimplerEnv | | | Real Robot | | |
|---|---|---|---|---|---|---|---|---|---|
| | Stage I | Stage II | Stage III | Stage I | Stage II | Stage III | Stage I | Stage II | Stage III |
| *Learning Rates* | | | | | | | | | |
| VLM LR | $1\times10^{-5}$ | $1\times10^{-5}$ | $1\times10^{-5}$ | $1\times10^{-5}$ | $1\times10^{-5}$ | $1.3\times10^{-5}$ | $1\times10^{-5}$ | $1\times10^{-5}$ | $1\times10^{-5}$ |
| DiT LR | $1\times10^{-4}$ | $1\times10^{-4}$ | $1\times10^{-4}$ | $1\times10^{-4}$ | $1\times10^{-4}$ | $1.3\times10^{-4}$ | $1\times10^{-4}$ | $1\times10^{-4}$ | $1\times10^{-4}$ |
| *Optimization Config* | | | | | | | | | |
| Action Horizon | 8 | 8 | 8 | 16 | 16 | 16 | 25 | 25 | 25 |
| Training Steps | 5k | 2k/2k/2k | 40k | 10k | 5k/5k/10k | 60k | 5k | 2k/2k/2k | 10k |
| Batch Size | 12 | 16 | 16 | 12 | 16 | 16 | 12 | 16 | 16 |
| Optimizer | AdamW | AdamW | AdamW | AdamW | AdamW | AdamW | AdamW | AdamW | AdamW |
| LR Scheduler | Cosine | Cosine | Cosine | Cosine | Cosine | Cosine | Cosine | Cosine | Cosine |
| Warm-up Ratio | 0.1 | 0.1 | 0.1 | 0.1 | 0.1 | 0.1 | 0.1 | 0.1 | 0.1 |
| *Loss Weights* | | | | | | | | | |
| Action Token Loss | 1.0 | 1.0 | – | 1.0 | 1.0 | – | 1.0 | 1.0 | – |
| Image Next Loss | 0.1 | 0.2 | – | 0.1 | 0.2 | – | 0.1 | 0.2 | – |
| CoT Loss | 1.0 | 1.0 | – | 1.0 | 1.0 | – | 1.0 | 1.0 | – |
| DiT Loss | – | – | 1.0 | – | – | 1.0 | – | – | 1.0 |

continuous-action flow matching head. For real-robot experiments, we train GR00T with an action chunk size of $K = 25$ and batch size 128 on a single H100 GPU. We follow the recommended optimization hyperparameters, using AdamW with learning rate $1 \times 10^{-4}$, weight decay $1 \times 10^{-5}$, and a warmup ratio of 0.05 of the total training steps. Consistent with the original setup, we freeze both the language model backbone and the vision tower, and fine-tune only the projector and diffusion policy head.

**ECoT (Zawalski et al., 2025).** We instantiate an ECoT-style baseline as an ablated variant of LaRA-VLA, retaining explicit textual CoT supervision while removing the action expert, visual CoT, and latent textual CoT components. All other model configurations and training hyperparameters are kept consistent with LaRA-VLA to ensure a fair comparison.

## B. Details of Data Pipeline

We argue that effective robotic manipulation relies on the organic integration of three tightly coupled components: subtask analysis, spatial grounding of target objects, and directional motion reasoning for the manipulator. Subtask decomposition enables long-horizon reasoning, spatial information localizes task-relevant objects, and motion reasoning translates high-level intent into executable control signals. However, existing data collection pipelines fail to jointly capture these components in a compact and task-centric manner (Zawalski et al., 2025; Sun et al., 2025b; Zhao et al., 2025b). For example, ECoT (Zawalski et al., 2025) annotates bounding boxes for all scene objects, leading to highly redundant supervision, while Emma-x (Sun et al., 2025b) lacks explicit target localization. To address this gap, we construct a fully automated annotation pipeline without human intervention, following an anchor-first, generate-later paradigm driven by semantic and temporal anchors, as shown in Figure 10.

**Subtask Annotation.** We extract semantic anchors using Qwen3-VL (Bai et al., 2025a) to identify the manipulated object from the first-frame image and the task instruction, which provides a textual reference for subsequent visual grounding. The prompt used for object identification is illustrated in Figure 11 Temporal anchors are obtained by segmenting robot trajectories into atomic manipulation stages (e.g., pre-grasp, grasp, move, and release) based on changes in the gripper state, with segment boundaries treated as keyframes. Conditioned on the instruction and the corresponding keyframes, Qwen3-VL generates high-level subtask descriptions for each segment. The prompt used for Subtask Annotation is illustrated in Figure 12.

**Target Object Bounding Boxes.** Guided by the semantic anchors, we perform open-vocabulary spatial grounding using GroundingDINO (Liu et al., 2024) and SAM3 (Carion et al., 2025) to obtain temporally consistent 2D bounding box trajectories of the target object. To be noted, for the Bridge dataset, we improve the robustness of our bounding box annotations by employing a multi-frame ensemble approach. We prompt SAM using GroundingDINO detections from 5 uniformly sampled anchor frames. The final trajectory is obtained by filtering spatial outliers and selecting the highest-

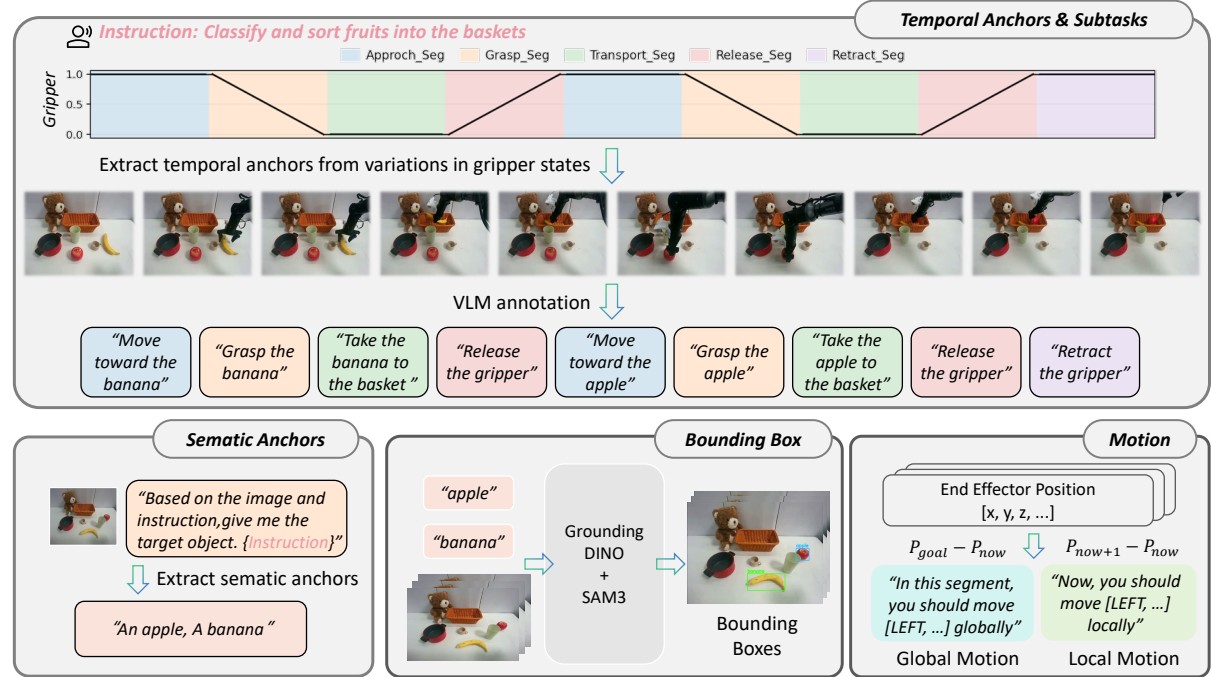

*Figure 10.* Data collection pipline.

confidence sequence. Furthermore, linear interpolation is applied to address tracking discontinuities, ensuring dense frame-wise supervision.

**Motion Reasoning.** Motion reasoning is derived from end-effector state trajectories by computing both global motion toward the segment goal and local instantaneous motion. These motion vectors are further mapped to directional descriptors and incorporated into the CoT annotations.

## C. Analysis of Real-World Experiments

Table 7 provides a subtask-level breakdown of real-world manipulation performance, revealing clear differences in temporal dependency across tasks. For tasks such as Put All Objects into the Basket, Put All Fruits into the Basket, and Stack Two Bowls, the two subtasks are largely decoupled and can be completed independently. This is reflected by the fact that failures in one subtask do not necessarily propagate to the other, and improvements in overall success rate are primarily driven by independent gains in individual subtasks. In contrast, Find the Block and Place It in the Basket exhibits strong sequential dependency between subtasks. Successful completion of the second subtask (placing the block) is contingent on correctly executing the first subtask (finding the block). As a result, errors in the initial subtask directly limit the achievable success rate of the subsequent subtask, leading to a tightly coupled failure mode. This structural dependency explains why improvements on this task require coherent reasoning and action execution across subtasks, rather than isolated subtask-level optimization. Notably, LaRA-VLA demonstrates consistent gains over GR00T N1.5 on this tightly coupled task, suggesting that its latent reasoning mechanism is particularly effective in maintaining cross-subtask coherence. By contrast, for tasks with weak subtask coupling, performance improvements can be largely attributed to better local action execution, and the benefits of reasoning primarily manifest as incremental gains.

*Table 7.* Subtask-level and overall success rates (%) on real-world robot tasks.

| Method | Put All Objects into the Basket | | | Sort All Fruits into the Basket | | | Find the Block and Place It in the Basket | | | Stack Two Bowls | | |
| --- | --- | --- | --- | --- | --- | --- | --- | --- | --- | --- | --- | --- |
| | Subtask 1 | Subtask 2 | Overall | Subtask 1 | Subtask 2 | Overall | Subtask 1 | Subtask 2 | Overall | Subtask 1 | Subtask 2 | Overall |
| ACT | 0.0 | 0.0 | 0.0 | 0.0 | 0.0 | 0.0 | 0.0 | 0.0 | 0.0 | 0.0 | 0.0 | 0.0 |
| GR00T N1.5 | 50.0 | 50.0 | **50.0** | 50.0 | 83.3 | 33.3 | 100.0 | 16.7 | 16.7 | 91.7 | 100.0 | 91.7 |
| LaRA-VLA | 50.0 | 41.7 | 41.6 | 66.7 | 75.0 | **50.0** | 100.0 | 33.3 | **33.3** | 100.0 | 100.0 | **100.0** |

```
```

You are helping a robot understand which object to manipulate.
You will receive:
1) A pre-grasp image.
2) The task instruction the robot must follow.

Instruction: ``{instruction}''

Your goal is to find the SINGLE primary object that the robot must interact with.
 - Read the instruction carefully to identify the nouns that describe the object(s) to
   be manipulated.
 - Prioritize what the instruction explicitly requests; ignore other objects unless
   they are essential context.
 - Ignore objects that the robot does not need to touch.
 - Describe the object with a short noun phrase (preferably including color if visible)
   but never mention any location context (e.g., no ``on stove'', ``in pot'', ``on
   left'').
 - The description must be ≤ 3 words, all lowercase, and purely object attributes.
 - List any other important objects mentioned in the instruction under
   secondary_objects (also ≤ 3 words, lowercase, no location mentions).

Output STRICT JSON with the following fields (no comments, no extra text):
``manipulated_object'':  string describing the main object (≤ 3 words, lowercase, no
location).
``secondary_objects'':  array of strings for other relevant instruction objects (each ≤
3 words, lowercase, no location).
If you are uncertain or there is no target object, set ``manipulated_object'' to the
noun mentioned in the instruction and secondary_objects to an empty array.
JSON:

```
```

*Figure 11.* Prompt for object identification.

# D. Additional Experiments

## D.1. Additional Analysis

**Effect of Action Pretraining.** We further study the effect of action supervision during the pretraining stages. Different from prior latent action methods, where latent actions are learned as intermediate variables to capture motion changes between frames, our latent reasoning variables are not latent actions. All three stages are directly supervised by ground-truth actions: Stages I and II use discrete action tokens to supervise explicit textual CoT and implicit latent textual reasoning, respectively, while Stage III adapts the model to downstream control with continuous action supervision. Table 8 reports an ablation on SimplerEnv, where we fix all other settings and evaluate checkpoints saved every 5k training steps from 35k to 60k. Action pretraining consistently improves performance across all checkpoints, increasing the average success rate from $60.4\%$ to $64.2\%$. This suggests that discrete action supervision during pretraining helps organize the latent reasoning space in a way that is better aligned with the robot's action space, thereby facilitating later continuous control learning.

*Table 8.* Effect of action pretraining on SimplerEnv. We compare models with and without discrete action supervision during the pretraining stages, while keeping all other settings fixed.

| Strategy | 35k | 40k | 45k | 50k | 55k | 60k | Avg. |
|---|---|---|---|---|---|---|---|
| w/o action pretraining | 55.2 | 59.4 | 64.6 | 62.5 | 60.4 | 60.4 | 60.4 |
| w/ action pretraining | **60.4** | **65.6** | **68.8** | **63.5** | **64.6** | **62.5** | **64.2** |

**Does CoT Supervision Help Without Inference-Time Reasoning?** Prior work, such as ECoT-lite (Chen et al., 2025c), suggests that CoT supervision may improve policy learning even when explicit CoT reasoning is not executed during inference. We therefore examine whether LaRA-VLA also benefits from such implicit reasoning effects. Specifically, we train the model with full latent reasoning supervision, but remove both the latent reasoning tokens and the `img_next` token from the prompt during inference, thereby disabling inference-time latent CoT and forcing the policy to directly generate

```
```

You are a robot manipulation expert.
Your goal is to describe what the robot is doing in this phase of a pick-and-place
task.

Global instruction:  ``{instruction}''
Internal segment label (for your reference only):  ``{segment_type}''

You are given two keyframes from this phase:  one near the beginning and one near the
end.
From the instruction and the images, you must:
1. Identify the single main object the robot is manipulating.  If the object cannot be
   identified, return ``unknown''.
2. Describe where this object is located or how it is situated in the scene
   (scene_context), e.g., ``on the table'', ``in the basket'', or ``in the box''.
3. Write a short natural-language subtask description for the current action phase,
   following the rules below:
    - The subtask MUST be a concrete action phrase (e.g., ``reach toward the blue cup'',
      ``carry the spoon toward the bowl'').
    - DO NOT simply repeat generic labels such as ``move_to_object'', ``move_to_goal'',
      ``grasp_object'', or ``place_object''.
    - The subtask should explicitly include the object name.  If the object cannot be
      identified, return the subtask as ``manipulate the object''.
Return STRICT JSON only (no extra text) with fields:
{
``object_name'':  ``string (e.g., 'blue cup')'',
``scene_context'':  ``string (e.g., 'inside the metal basket')'',
``subtask'':  ``string (natural language, must include the object)''
}

```
```

*Figure 12.* Prompt for subtask description generation.

actions without attending to explicit CoT-related tokens.

Table 9 reports the results on four SimplerEnv tasks. Training with CoT supervision but disabling CoT at inference still improves the average success rate from 55.2% to 61.4%, compared with the model trained and evaluated without CoT. This indicates that CoT supervision does not only provide useful intermediate tokens at inference time, but also helps shape the underlying image-instruction representation space during training. Nevertheless, enabling CoT reasoning at inference further improves the average success rate to 68.7%, showing that explicit latent reasoning remains beneficial beyond the implicit representation-level gains induced by CoT supervision.

*Table 9.* Effect of CoT supervision and inference-time reasoning on SimplerEnv. We compare models trained with or without CoT supervision and evaluate whether CoT-related tokens are used during inference. Training with CoT supervision improves performance even when CoT tokens are removed at inference time, while using latent reasoning during inference yields the best overall result.

| Setting | Put Spoon | Put Carrot | Stack Block | Put Eggplant | Avg. |
|---|---|---|---|---|---|
| Train w/o CoT, Infer w/o CoT | 79.2 | 37.5 | 16.7 | 87.5 | 55.2 |
| Train w/ CoT, Infer w/o CoT | 58.3 | 87.5 | 16.7 | 83.3 | 61.4 |
| Train w/ CoT, Infer w/ CoT | 95.8 | 62.5 | 25.0 | 91.7 | **68.7** |

**Effect of EMA on Latent Stability.** We further analyze the role of the EMA target encoder in stabilizing latent representation learning. EMA is introduced to provide a slowly evolving target for visual-latent supervision, which can reduce representation drift during the curriculum transition from explicit CoT to latent reasoning. To evaluate its effect, we compare models trained with and without EMA.

First, we measure the feature-space rank of the <img_next> latents as an indicator of representation diversity. While we do not observe severe latent collapse without EMA, removing EMA clearly reduces feature diversity: the effective rank

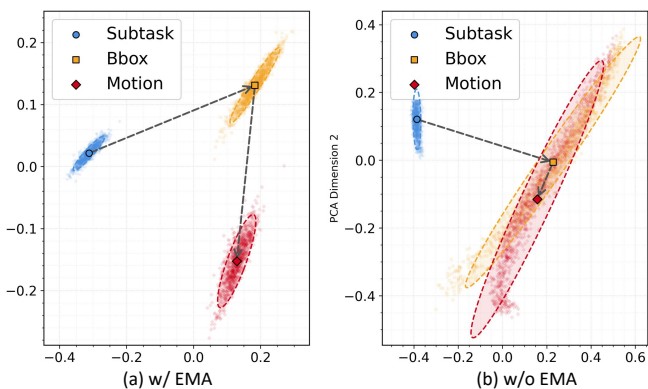

*Figure 13.* Effect of EMA on latent text token distributions. Without EMA, bounding-box and motion-related latent tokens exhibit stronger overlap, indicating greater semantic entanglement. EMA improves the separation of different reasoning components and stabilizes latent CoT learning.

decreases from 6.76 with EMA to 5.10 without EMA. Second, we visualize the latent text tokens in Figure 13. Without EMA, the latent tokens associated with bounding-box grounding and motion reasoning show stronger overlap, indicating increased semantic entanglement between different reasoning components. In contrast, EMA leads to better separated latent clusters, suggesting that the target encoder helps preserve more structured latent representations.

These results indicate that EMA is not strictly necessary to avoid immediate collapse, but it improves latent diversity and semantic separation, making latent CoT learning more stable. We therefore treat EMA as a stabilizing training component rather than an independent contribution.

