# OpenReview forum: "Latent Reasoning VLA: Latent Thinking and Prediction for Vision-Language-Action Models"
_ICML.cc/2026/Conference — ICML 2026 regular_

### Official Review · Reviewer_Urck · 2026-02-17

**Soundness:** 4
**Presentation:** 4
**Significance:** 3
**Originality:** 3
**Overall Recommendation:** 5
**Confidence:** 4

**Summary:**

The contribution of this paper is a latent chain of though method for a VLA which combines both textual and visual inputs and supervision. The method uses a curriculum to move from explicit supervision to latent reasoning, by gradually masking ground truth textual and visual supervision signals, and replacing them with the learned representations. Action prediction between two visual state observations is also employed to shape the latent space. The work also introduces two adapted datasets, annotated with the training labels for the chain of thought supervision - semantic anchors such as the manipulated objects, temporal manipulation stages and motion reasoning, subtask descriptions, and bounding boxes.

**Compliance With Llm Reviewing Policy:**

Affirmed.

**Final Justification:**

I am happy to recommend acceptance for this paper.

**Key Questions For Authors:**

1. Why were ACT and GR00T1.5 selected as baselines? It would be great to have discussion of this in the main text.
2. Was the exponential moving average essential to your approach? It is mentioned in the contribution but does not appear to be evaluated.
3. Well done for including Fast-ThinkAct in the paper given its release 14 days before the submission deadline. However, since it is included I must ask: can you explain the key differences between your methods? This discussion would be very valuable in the literature review.

**Limitations:**

Yes

**Strengths And Weaknesses:**

I think the authors for their submission, which I greatly enjoyed reading.

The strengths of this work are:
* Interesting and well reasoned method, with great results
* Good selection of baselines, especially Fast-ThinkAct was excellent to include
* Clear presentation of text, tables, and figures
* Great evaluation, especially using real world experiments
* Contribution of their training data and annotated training sets
* The appendix is clear and helpful, especially for the training datasets

The weaknesses of this work are:
* The closest related work appears to be Fast-ThinkAct, which was included as the only baseline which also does latent reasoning. This work also uses textual and visual information as part of the latent chain of thought. However, this work was not discussed in the literature review. Their work was only put onto Arxiv on 14th January 2026, so it demonstrates high integrity and rigour that it was included as a baseline. However, since it is included in the paper, it should be discussed in the literature review. It should be explained how your contribution is different from theirs, and the benefits of your method. It should also be added to Table 1.
* Similarly to the previous point, Fast-ThinkAct is included in the latency analysis in Figure 7, but not discussed in the 'Inference Efficiency' text, which only refers to 'explicit CoT methods' rather than the latent CoT of Fast-ThinkAct. I appreciate your inclusion of this baseline, and I feel it makes the paper stronger overall. I would recommend you reformat this discussion to illustrate how your method is superior to theirs, and make it clearer that they are using latent chain of thought.
* It would be valuable to explain to readers why ACT and GR00T1.5 were selected as real world baselines. This explanation should also answer whether these models use chain of thought, and if so what type? It is helpful for the reader to have a consistent criteria for comparison and evaluation across both the simulation and real world experiments. Similarly, Figure 5 gives the real world results, but does not indicate what kind of chain of thought is used, whereas all previous evaluation tables have focused on this element. The caption for Figure 5 is extremely brief, but could contain this additional context.
* The exponential moving average is mentioned as critical to the method in the contribution section, and referenced several time during the methods. However, it is not present during the evaluation. Instead, there is analysis indicating no latent collapse. This analysis is good, but a key question must be answered: is there latent collapse without the exponential moving average? This could be added to the appendix, or space-permitting in the main body, but I feel this is an essential addition to the paper as it is mentioned in the contribution. If the EMA is not essential to your method, I would suggest it be removed from the contribution, and a brief discussion of its benefits (e.g. slight stability improvements) simply be discussed or shown in the appendix.

---

> ### Author Rebuttal · Authors · 2026-03-31
>
> Thanks for your affirmative review and we address your questions as follows.
>
> ## Please open [link to *Anonymous GitHub repository*](https://anonymous.4open.science/r/Rebuttal-975E/README.md) for easy reference.
>
> ### 1. Weakness 1 and Question 3 on clarity of Fast-Thinkact: Fast-ThinkAct is the closest related work and should be discussed in the literature review. The paper should clarify how the proposed method differs from it, what advantages it offers, and include it in Table 1.
>
> We thank the reviewer for raising this comparison. Although both Fast-ThinkAct and LaRA-VLA replace part of explicit CoT with latent reasoning, they differ in three key aspects.
> (1) **Training paradigm.** Fast-ThinkAct follows a teacher-student distillation pipeline, whereas LaRA-VLA adopts a progressive curriculum that gradually transitions from explicit CoT to latent reasoning.
> (2) **Supervision of text latent reasoning.** Fast-ThinkAct uses an additional verbalizer LLM to decode latent representations back into text for explicit language-space supervision. In contrast, LaRA-VLA does not impose such direct reconstruction supervision; its latent text reasoning is constrained implicitly through downstream visual reasoning and discrete action supervision.
> (3) **Form of visual reasoning.** Fast-ThinkAct uses visual traces as auxiliary discrete visual information, which in our taxonomy is closer to visually grounded textual CoT. By contrast, LaRA-VLA uses future image tokens as visual CoT, thereby internalizing both textual reasoning and visual prediction into continuous latent representations.
>
> 2. We will add a discussion of Fast-ThinkAct in the **introduction and related work** to clarify its differences from LaRA-VLA, and we will revise Table 1 in the main text accordingly, as shown below.
>
> |Method|Venue|Text CoT Have|Text CoT Form|Visual CoT Have|Visual CoT Align Form|Action Expert|Action Form|
> |-|-|-|-|-|-|-|-|
> |Fast-ThinkAct|CVPR 2026|√|Discrete Token & Continuous Latent|×|-|√|Continuous|
>
> ### 2. Weakness 2 on latency experiments of Fast-Thinkact: Fast-ThinkAct is included in Figure 7, but the efficiency discussion does not clearly address it as a latent CoT baseline. The paper should explicitly compare the two methods and clarify the efficiency advantage of the proposed method.
>
> One reason why Fast-ThinkAct is slower than LaRA-VLA is that it uses more latent text tokens during inference, specifically 6 versus 3 in our method. A second possible factor is the number of visual trace tokens, which is not explicitly reported in the Fast-ThinkAct paper but may exceed the 16 visual latent tokens used in LaRA-VLA. A third possibility is that its action model is larger than ours, as it adopts an RDT-based action module. Since Fast-ThinkAct has not been open-sourced, these explanations are necessarily based on our reading of the paper rather than direct implementation-level verification. We will clarify these factors in the experimental section.
>
> ### 3. Weakness 3 and Question 1 on real-world baselines: It would be helpful to clarify why ACT and GR00T N1.5 were chosen as real-world baselines, including whether they use CoT and what type. Figure 5 and its caption could also provide this context for a more consistent comparison.
>
> (1) We select ACT and GR00T N1.5 because they are widely adopted real-world baselines, and LaRA-VLA without CoT-related components is architecturally closest to GR00T N1.5. Neither baseline uses CoT.
> (2) To further ensure fair comparison, we additionally include an explicit text CoT baseline, ECoT, yielding a more complete real-world comparison across no-CoT, explicit CoT, and latent CoT settings. Specifically, we train ECoT on the same 4B backbone and dataset as LaRA-VLA, with results shown below:
>
> |Method|Put all objects into the basket|Put all fruits into the basket|Find the block and place it in the basket|Stack two bowls|Avg|
> |-|:-:|:-:|:-:|:-:|:-:|
> |ECoT|16.7|8.3|0|58.3|20.8|
> |LaRA-VLA|41.6|50.0|33.3|100.0|56.2|
>
> ### 4. Weakness 4 and Question 2 on EMA: EMA is presented as important, but its effect is not directly evaluated. The key question is whether latent collapse occurs without EMA. If not essential, it should be removed from the contributions and discussed more modestly.
>
> We employ EMA to preserve feature diversity and stabilize latent text CoT learning. To evaluate its effect, we compare models trained with and without EMA.
> (1) We first analyze the latent space of `<img_next>`. We do not observe obvious feature collapse without EMA, but feature diversity is reduced: the feature-space rank drops from 6.76 with EMA to 5.10 without EMA.
> (2) We further analyze the latent text tokens, as visualized in **Figure 4 of the Anonymous GitHub repository**. Without EMA, the latent tokens corresponding to bounding boxes and motion exhibit substantial overlap, indicating a stronger degree of entanglement. This harms the semantic separation of text latent CoT and increases the risk of latent collapse.

---

> > ### Author Rebuttal · Reviewer_Urck · 2026-04-03
> >
> > I thank the authors for their responses, which address my concerns.

---

> > > ### Author Response · Authors · 2026-04-04
> > >
> > > Thank you very much for your positive update. We sincerely appreciate your time and thoughtful feedback, and we are glad that our rebuttal was able to address your concerns.

---

### Official Review · Reviewer_1Qpg · 2026-03-04

**Soundness:** 3
**Presentation:** 3
**Significance:** 2
**Originality:** 3
**Overall Recommendation:** 4
**Confidence:** 4

**Summary:**

This paper proposes Latent Reasoning VLA (LaRA-VLA), a vision-language-action framework that internalizes chain-of-thought (CoT) reasoning into continuous latent representations to avoid explicit CoT generation at inference time. The method uses a three-stage curriculum: (i) explicit multimodal CoT fine-tuning with future-visual-latent prediction and action supervision, (ii) progressively replacing discrete CoT tokens with learnable latent “thinking” representations, and (iii) conditioning a diffusion/flow-matching action expert on the learned latent context for continuous control. Experimental results show that LaRA-VLA achieves strong benchmark results on LIBERO and SimplerEnv-WidowX, along with large inference-latency reductions.

**Compliance With Llm Reviewing Policy:**

Affirmed.

**Final Justification:**

My concerns have been addressed, and I will maintain my original score.

**Key Questions For Authors:**

LaRA-VLA requires multiple models for intermediate annotation. How much additional annotation and training overhead does this introduce? Does each dataset require this extra processing? If so, could this pose challenges for real-world deployment and application?

**Limitations:**

yes

**Strengths And Weaknesses:**

## Strength
1. The high-level motivation and idea is easy to follow, and the three-stage pipeline is clearly conveyed.
2. The staged training objective and architecture choices are well-motivated for reducing auto-regressive token generation overhead while preserving multi-step computation.
3. The real-world experiments explicitly focus on long-horizon tasks and provide subtask-level analysis, which is valuable for understanding temporal dependency and failure propagation.

## Weakness
1. The latency figure compares models with different parameter scales (e.g., ThinkAct-7B / ECOT-7B vs LaRA-VLA-4B), which confounds whether gains come from “latent reasoning” vs simply smaller models and/or different implementations. A controlled comparison (same backbone size, same hardware, same decoding settings) would strengthen the claim.
2. ACT scores 0% across all real-world tasks, which is unusually low and raises questions about tuning, data preprocessing, or evaluation protocol compatibility. Additionally, LaRA-VLA does not uniformly exceed GR00T N1.5 per-task: e.g., “Put All Objects into the Basket” overall success is 41.6 for LaRA-VLA vs 50.0 for GR00T N1.5. This contradicts the text’s “consistently outperforms … across all four tasks” phrasing and should be corrected/qualified.
3. The automated CoT annotation pipeline relies heavily on strong external models (Qwen3-VL, GroundingDINO, SAM3). The paper does not quantify label noise or sensitivity to annotation errors; since these labels supervise intermediate reasoning, it is important to assess robustness when grounding/segmentation fails.

---

> ### Author Rebuttal · Authors · 2026-03-31
>
> Thanks for your affirmative review of this paper! We address your questions as follows.
>
> ### 1. Weaknesses 1: the latency figure compares models with different parameter scales (e.g., ThinkAct-7B / ECOT-7B vs LaRA-VLA-4B), which confounds whether gains come from “latent reasoning” vs simply smaller models and/or different implementations. A controlled comparison (same backbone size, same hardware, same decoding settings) would strengthen the claim.
>
> We thank the reviewer for this well-motivated question. To address this concern, we implement a controlled baseline ablation, denoted as ECoT-4B, on top of our LaRA-VLA framework. This baseline uses the same 4B-parameter backbone, is trained on the same CoT training data, and is evaluated on the same NVIDIA A100 GPU.
>
> |Method|Latency(ms)|
> |-|:-:|
> |ECoT-4B|1400|
> |LaRa-VLA-4B|135|
>
> The result shows an approximately 10$\times$ reduction in inference latency at the same model scale. This substantial improvement indicates that the efficiency gain does not simply come from model size, but from our latent reasoning formulation, which avoids explicit long token-by-token textual decoding during inference. We will update Figure 7 in main text and the corresponding discussion in the revised manuscript.
>
> ### 2. Weaknesses 2 on experiments and statements: ACT scores 0% across all real-world tasks, which is unusually low and raises questions about tuning, data preprocessing, or evaluation protocol compatibility. Additionally, LaRA-VLA does not uniformly exceed GR00T N1.5 per-task: e.g., “Put All Objects into the Basket” overall success is 41.6 for LaRA-VLA vs 50.0 for GR00T N1.5. This contradicts the text’s “consistently outperforms ... across all four tasks” phrasing and should be corrected/qualifie.
>
> (1) The ACT baseline in our experiments is implemented using the LeRobot version, with full configuration details provided in Appendix Section A.2. To further address the reviewer’s concern, we additionally include an explicit text CoT baseline, ECoT, thereby providing a more complete real-world comparison across no-CoT, explicit CoT, and latent CoT settings, as shown in the table below:
>
> |Method|Put all objects into the basket|Put all fruits into the basket|Find the block and place it in the basket|Stack two bowls|Avg|
> |-|:-:|:-:|:-:|:-:|:-:|
> |ECoT|16.7|8.3|0|58.3|20.8|
> |LaRA-VLA|41.6|50.0|33.3|100.0|56.2|
>
> (2) We will revise the statement in the main text by replacing “consistently outperforms ...” with “achieves better overall performance than ...” to make the claim more precise and better supported by the empirical results.
>
> ### 3. Weaknesses 3 on CoT datasets: the automated CoT annotation pipeline relies on strong external models, but the paper does not quantify label noise or sensitivity to annotation errors. Since these labels supervise intermediate reasoning, robustness to grounding or segmentation failures should be assessed.
>
> We clarify that potential annotation errors are mainly limited to bounding box generation. In contrast, subtask reasoning is produced by Qwen3-VL under explicit instruction guidance, while motion reasoning is grounded in reliable metadata.
> For bounding box annotation, we employ a robust pipeline with multiple prompts, SAM output merging, and strict filtering thresholds. Since our model maps bounding boxes into latent representations without an explicit decoding step, grounding failures cannot always be directly observed. To quantify annotation quality, we therefore manually select 1000 samples from both the LIBERO and Bridge datasets using an IoU > 0.75 criterion. We find that 948 annotations are correct on LIBERO and 915 on Bridge, which supports the effectiveness of our pipeline.
> Moreover, the architecture of LaRA-VLA itself provides an additional degree of robustness. By internalizing reasoning in a continuous latent space, the model is encouraged to capture consistent task-relevant structure rather than overfit to explicit annotation noise.
>
> ### 4. Question 1: LaRA-VLA requires multiple models for intermediate annotation. (1) How much additional annotation and training overhead does this introduce? (2) Does each dataset require this extra processing? If so, could this pose challenges for real-world deployment and application?
>
> (1) Our annotation pipeline is automated and performed entirely offline. For the BridgeV2 dataset, the complete process takes approximately 1.5 days on an H100 GPU, with the majority of the runtime spent on SAM-based object tracking. Although the current training pipeline relies on these reasoning annotations, this cost is incurred only once during preprocessing and is justified by the resulting gains in task success rate and inference efficiency.
> (2) Because the entire pipeline is fully automated and requires no additional manual processing, it can be extended to new datasets with minimal human effort.
> We will include these implementation details and a discussion of the associated overhead in the revised manuscript.

---

> > ### Author Rebuttal · Reviewer_1Qpg · 2026-04-03
> >
> > Thank you for the authors’ response. My concerns have been addressed, and I have no further questions.

---

> > > ### Author Response · Authors · 2026-04-04
> > >
> > > Thank you for the positive update. Since the acknowledgement is marked as “Fully resolved” and you mentioned that your concerns have been addressed with no further questions, I was wondering whether there are any remaining considerations affecting the score.

---

### Official Review · Reviewer_cete · 2026-03-11

**Soundness:** 3
**Presentation:** 3
**Significance:** 3
**Originality:** 3
**Overall Recommendation:** 4
**Confidence:** 4

**Summary:**

This paper proposes a training framework that performs both textual and visual chain-of-thought (CoT) reasoning at the level of continuous latent representations rather than discrete tokens, thereby improving inference efficiency while simultaneously enhancing VLA performance.

**Compliance With Llm Reviewing Policy:**

Affirmed.

**Final Justification:**

The authors have addressed my concerns, so I maintain my initial positive score.

**Key Questions For Authors:**

1. In Table 4, what is the performance when only Latent Vis-CoT is applied?

2. In prior work such as ECoT-lite, it has been shown that training with CoT supervision can improve performance even when CoT reasoning is not executed at inference time. In this regard, I am curious how the proposed model would perform if it were trained with CoT supervision but directly generated actions without performing CoT reasoning during inference.

**Limitations:**

yes

**Strengths And Weaknesses:**

Strengths
1. Chain-of-thought (CoT) reasoning has been shown to be effective in the VLA literature, but it has rarely been used in practice due to its high inference cost. This paper proposes a method that significantly reduces this overhead. While the reported generation latency of 135 ms is still relatively slow, it falls within a practically usable range. This represents a meaningful step forward for the field.
2. The results on LIBERO, SimplerEnv, and real-world experiments are impressive.
3. The ablation studies and analyses are appropriate and provide sufficient insights into which components of the model contribute to the performance improvements.


Weaknesses
1. One of the commonly cited benefits of incorporating CoT reasoning is improved generalization performance. However, the benchmarks used in this paper, LIBERO and SimplerEnv, make it difficult to assess the model's generalization ability, and the real-world experimental setup has a similar limitation. It would be valuable to evaluate the model under OOD settings to observe how performance changes in unseen environments.

2. The paper does not demonstrate whether the proposed approach improves performance in an action pretraining setup. In particular, when latent actions are learned from pretraining data from a different domain (e.g., BridgeV2), it remains unclear whether this knowledge effectively transfers to new domains during fine-tuning.

---

> ### Author Rebuttal · Authors · 2026-03-31
>
> ## Please open [link to *Anonymous GitHub repository*](https://anonymous.4open.science/r/Rebuttal-975E/README.md) for easy reference.
>
> ### 1. Weakness 1 on generalization ability: OOD generalization is a common claim of CoT reasoning, but the current evaluations do not clearly assess it. Testing under unseen or corrupted conditions would better reveal generalization.
>
> We thank the reviewer for this helpful suggestion, which improves the completeness of our evaluation and provides additional insight into generalization under OOD settings. To address this, we conduct additional experiments on the LIBERO and SimplerEnv benchmarks using two common perceptual shifts, Gaussian noise and Gaussian blur, with detailed configurations provided in **Figure 2 of the Anonymous GitHub repository**.
> We compare LaRA-VLA with Qwen-GR00T, which can be viewed as LaRA-VLA without all CoT reasoning, under the same perturbation settings, as shown in the **Table below**. While both models degrade under corruption, LaRA-VLA consistently retains higher success rates and stronger stability. These results further support the effectiveness of latent reasoning in preserving task-relevant structure under perceptual corruption.
> |Benchmark|Method|Blur Gaussian (High)|Blur Gaussian (Low)| Gaussian Noise (High)|Gaussian Noise (Low)|
> |-|-|:-:|:-:|:-:|:-:|
> |LIBERO|Qwen-GR00T|30.0|76.0|55.7|87.9|
> |LIBERO|LaRA-VLA|**42.9**|**88.2**|**76.0**|**97.0**|
> |Simpler|Qwen-GR00T|13.5|35.4|8.3|22.9|
> |Simpler|LaRA-VLA|**56.3**|**62.5**|**22.9**|**31.2**|
>
> ### 2. Weakness 2 on action pretraining: the paper does not show whether the method improves action pretraining or whether latent actions learned from one domain (e.g., BridgeV2) transfer effectively to new domains during fine-tuning.
>
> (1) We believe the reviewer may be conflating our method with prior latent action approaches. In prior work, latent actions denote learned intermediate variables that capture discretizable motion changes between the current frame and a future frame [1,2]. Our method is different: none of the three training stages uses latent actions. Instead, all stages are supervised by ground-truth actions, with discrete action tokens used in Stages I and II, and continuous actions in Stage III.
> (2) To clarify the role of action pretraining in Stages I and II, we provide an ablation study on Simpler in the **table below**. All other settings are fixed, and checkpoints are saved every 5k steps from 35k onward. The results show that discrete action supervision during pretraining helps align the latent space with the robot’s discrete action space, consistent with [3].
>
> |Strategy\step|35k|40k|45k|50k|55k|60k|
> |-|:-:|:-:|:-:|:-:|:-:|:-:|
> |w/o action pretraining|55.2|59.4|64.6|62.5|60.4|60.4|
> |w/ action pretraining|60.4|65.6|68.8|63.5|64.6|62.5|
>
> ### 3. Question 1 on ablation: in Table 4, what is the performance when only Latent Vis-CoT is applied?
>
> To isolate the contribution of visual reasoning, we conduct an additional ablation by training a variant that uses only Latent Vis-CoT (i.e., the `img_next` token) together with action tokens. For this variant, we simplify the curriculum into two stages: (i) joint supervision of future-image prediction and action tokens to ground the latent space in visual dynamics, and (ii) standard policy fine-tuning based on the pretrained latent representations. All hyperparameter settings are kept identical to those of the main LaRA-VLA model. As shown in the **Table below**, the results further support the effectiveness of both reasoning modalities.
> |Text-CoT|Latent Text-CoT|Latent Vis-CoT|Avg|
> |:-:|:-:|:-:|:-:|
> |x|x|x|55.21|
> |√|x|x|58.33|
> |x|√|x|64.58|
> |x|x|√|63.50|
> |x|√|√|68.75|
>
> ### 4. Question 2 on discussion: As shown in ECoT-lite, CoT supervision can help even without CoT at inference. It would be valuable to test this setting for the proposed model.
>
> To answer this question, we conduct an additional experiment in which the model is trained with the full latent reasoning strategy, while both the latent reasoning tokens and the `img_next` token are removed from the prompt during inference, thereby disabling explicit CoT. The results are reported in the **table below**. Interestingly, the model still retains non-trivial manipulation performance, similar to the ECoT-lite observation. We hypothesize that CoT supervision reshapes the joint image-instruction representation during training, inducing an implicit reasoning capability that continues to support action generation at test time.
> |Method|Carrot|Spoon|Cube|Eggplant|Avg|
> |-|:-:|:-:|:-:|:-:|:-:|
> |train w/o CoT, inference w/o CoT|79.2|37.5|16.7|87.5|55.2|
> |train w/ CoT, inference w/o CoT|58.3|87.5|16.7|83.3|61.4|
> |train w/ CoT, inference w/ CoT |95.8|62.5|25.0|91.7|68.7|
>
> [1] Latent action pretraining from videos. ICLR 2025.
> [2] Univla: Learning to act anywhere with task-centric latent actions. RSS 2025.
> [3] $π_{0.5}$: a Vision-Language-Action Model with Open-World Generalization. CoRL 2025.

---

> > ### Author Rebuttal · Reviewer_cete · 2026-04-03
> >
> > Thanks to the authors for their detailed response. All my concerns have been resolved.

---

> > > ### Author Response · Authors · 2026-04-04
> > >
> > > Thank you for the encouraging update and for stating that all concerns have been resolved. Since the acknowledgement is marked as “Fully resolved,” I was wondering whether there are any remaining considerations affecting the score.

---

### Official Review · Reviewer_eoR9 · 2026-03-11

**Soundness:** 3
**Presentation:** 3
**Significance:** 3
**Originality:** 3
**Overall Recommendation:** 4
**Confidence:** 3

**Summary:**

Paper proposes LaRA-VLA a latent-reasoning paradigm for Vision–Language–Action models to represent text or image chain-of-thought generation with structured intermediate computation in continuous latent space. The main motivation presented is inference overhead of text CoT (KV-cache growth, memory, latency) and the representational mismatch introduced by discrete visual tokens in embodied control, the method trains a unified VLA model via curriculum learning.

Also curated LIBERO-LaRA and Bridge-LaRA datasets spanning simulated and real-robot long-horizon manipulation and implement the model with Qwen3-VL as backbone.

Training is done in 3 different stages. Explicit CoT fine-tuning, where visual latent learning update of parameters via EMA. Stage 2 is to replace discrete CoT Tokens via a curriculum based approach where subsets of CoT tokens are masked out and replace them with learnable latent representations. Finally. action pred model is trained via flow-matching.

Attention mechanism introduced for the 3 stage training paradigm to regulate the cross-token information flow across text, 2 images(current and future) and action tokens.

**Compliance With Llm Reviewing Policy:**

Affirmed.

**Key Questions For Authors:**

1. please share any information on the train wall times for 3 stage training
2. How stable or unstable was training with these three stages and with new attention introduced to incorporate different modalities.

**Limitations:**

yes

**Strengths And Weaknesses:**

S:
1. beyond simulation, including long-horizon real-world manipulation with a concrete robot setup and compares two different baselines ACT and Groot.
2. presents inference time comparisons and improvements/efficiency
3. useful ablations
4. Overall,compares with broad set of baselines.
W:
1. paper does not present any wall-times on the three stages of training, this is important as one of the initial claims is training is expensive as more CoT tokens are added and states training efficiency as main drwaback that needs to be addressed.
2. stability of the latent space to perturbation, results present no latent collapse via clustering in a qualitative way but no robustness to perturbations such as distribution shifts or change in any parameter

---

> ### Author Rebuttal · Authors · 2026-03-31
>
> Thanks for your affirmative review of this paper! We address your questions as follows.
>
> ## Please open [link to *Anonymous GitHub repository*](https://anonymous.4open.science/r/Rebuttal-975E/README.md) for easy reference.
>
> ### 1. Weakness 1 on the clarity of the training time across the three stages: paper does not present any wall-times on the three stages of training, this is important as one of the initial claims is training is expensive as more CoT tokens are added and states training efficiency as main drwaback that needs to be addressed.
>
> We thank the reviewer for raising this important point. To substantiate our claim that training cost increases with the number of latent tokens, we report empirical wall-clock measurements from Bridge dataset experiments conducted on 8×H100 GPUs. The results show a near-linear increase in per-step forward latency during Stage 2, rising from 0.60 s with 1 latent token to 0.85 s with 2 tokens and 1.25 s with 3 tokens. This confirms that increasing the number of latent tokens directly increases overall training time.
> To balance reasoning capacity and computational efficiency, our method compresses essential information into a minimal number of latent tokens, with one reasoning step represented by one latent token. Under this design, Stage 1 requires approximately 1.7 hours, Stage 2 requires 6.2 hours, and Stage 3 policy learning takes 23 hours, resulting in about 31 hours of total training time. We will include these wall-clock statistics and the corresponding analysis in the revised manuscript.
>
> ||Stage I|Stage II-step 1|Stage II-step 2|Stage II-step 3|Stage III|
> |-|-|-|-|-|-|
> |Training Step|10k|5k|5k|10k|60k|
> |Training Time (H100 Hours)|13.6|9.6|12|27.8|186.4|
> |One Forward Pass (H100 Seconds)|0.6|0.85|1.1|1.25|1.4|
>
> ### 2. Weakness 2 on experiments of latent stability: stability of the latent space to perturbation, results present no latent collapse via clustering in a qualitative way but no robustness to perturbations such as distribution shifts or change in any parameter.
>
> To further assess the robustness of the learned latent representations, we conduct additional stress tests on the SimplerEnv and LIBERO benchmarks using two common visual perturbations, Gaussian noise and Gaussian blur, each with two severity levels. We analyze both the resulting changes in latent-space distributions and the corresponding task success rates.
> As shown in **Figure 1 of the Anonymous GitHub repository**, the latent reasoning tokens remain semantically coherent under corrupted inputs, with only modest shifts in the latent space. We further compare LaRA-VLA with the baseline Qwen-GR00T, which can be viewed as LaRA-VLA without all CoT reasoning, under identical perturbation settings; detailed noise configurations are provided in **Figure 2 of the Anonymous GitHub repository**. As shown in the **Table below**, although both models degrade under visual corruption, LaRA-VLA consistently retains substantially higher success rates across all settings. These results provide further evidence that latent reasoning improves robustness to input noise. We will include these additional results and the corresponding analysis in the revised manuscript.
>
> |Benchmark|Method|Blur Gaussian (High)|Blur Gaussian (Low)| Gaussian Noise (High)|Gaussian Noise (Low)|
> |-|-|:-:|:-:|:-:|:-:|
> |LIBERO|Qwen-GR00T|30.0|76.0|55.7|87.9|
> |LIBERO|LaRA-VLA|**42.9**|**88.2**|**76.0**|**97.0**|
> |Simpler|Qwen-GR00T|13.5|35.4|8.3|22.9|
> |Simpler|LaRA-VLA|**56.3**|**62.5**|**22.9**|**31.2**|
>
> ### 3. Question 1 on the clarity of the training time: please share any information on the train wall times for 3 stage training.
>
> Thank you for the question. We have addressed this point in our **response to Weakness 1**, where we provide clarification on the wall-clock training time of the three training stages.
>
> ### 4. Question 2 on training stability: how stable or unstable was training with these three stages and with new attention introduced to incorporate different modalities.
>
> As shown in **Figure 3 of the Anonymous GitHub repository**, the training loss curves exhibit stable and consistent convergence across all three stages. The stability of Stage I is expected, while the stable optimization in Stage II is enabled by our progressive curriculum design, where the explicit grounding learned in Stage I provides a semantic prior for the transition to latent reasoning. Stage III also shows stable adaptation behavior, as evidenced by its smooth training trajectory in the same figure.
> Notably, the EMA mechanism we introduce also helps stabilize the training of latent text reasoning. As shown in **Figure 4 of the Anonymous GitHub repository**, adding EMA leads to semantically coherent latent clusters, rather than latent collapse.
> We will include these results, together with the corresponding analysis, in the revised paper to further substantiate this training stability.

---

> > ### Author Rebuttal · Reviewer_eoR9 · 2026-04-02
> >
> > my questions and concerns were answered adequately

---

> > > ### Author Response · Authors · 2026-04-04
> > >
> > > Thank you for the update and for indicating that your concerns have been fully resolved. May I ask whether there are any remaining factors, beyond the rebuttal points, that are still affecting the overall score?

---

### Decision · Program_Chairs · 2026-04-30

**Decision:**

Accept (regular)

**Comment:**

This paper presents a unified framework that incorporates multi-modal CoT reasoning into continuous latent representations for embodied action. The authors also construct two structured CoT datasets, LIBERO-LaRA and Bridge-LaRA, and evaluate LaRA-VLA across simulation benchmarks and long-horizon real-robot manipulation tasks, where impressive results are presented.

All reviewers are positive on this work, ranging from weak accept to accept. Reviewers' concerns are also well addressed during rebuttal. I agree with reviewers and recommend accepting.